# Minimum Clearance Distance in Fall Arrest Systems with Energy Absorber Lanyards

**DOI:** 10.3390/ijerph18115823

**Published:** 2021-05-28

**Authors:** Elena Ángela Carrión, Belén Ferrer, Juan Francisco Monge, Pedro Ignacio Saez, Juan Carlos Pomares, Antonio González

**Affiliations:** 1Building & Urban Development Department, University of Alicante, 03690 Alicante, Spain; pi.saez@ua.es; 2Civil Engineering Department, University of Alicante, 03690 Alicante, Spain; belen.ferrer@ua.es (B.F.); jc.pomares@ua.es (J.C.P.); antonio.gonzalez@ua.es (A.G.); 3Center of Operations Research, Miguel Hernandez University of Elche, 03202 Elche, Spain; monge@umh.es

**Keywords:** energy absorber lanyard, fall arrest systems, dynamic performance test, clearance distance, elastic and plastic deformation, high-speed camera

## Abstract

Accidents due to falls from height are one of the main causes of death in workplaces. Fall arrest systems (FAS) are designed to arrest the fall safely without injuring the accident victim. Their main mission is to restrain the body so as to prevent it from hitting the ground, generating forces and accelerations in the arrest process that are not harmful. A lack of empirical evidence and scant technical information provided by manufacturers regarding minimum clearance distance (MCD) below anchorage makes it necessary to study the safety distances required in the use of Energy Absorber Lanyards (EAL). This paper aims to determine the MCD below anchorage to arrest a fall using an EAL without hitting the ground. The real deformation of EAL when stopping a fall is studied. Ten EAL models distributed internationally by six manufacturers were chosen. Dynamic behavior tests were performed on the samples. Using image processing the total elongation of the equipment (elastic plus plastic) is obtained. The main conclusions are that maximum plastic elongation could be reduced by 29%. The method described in EN 355: 2002 underestimates elongation by up to 70% for some equipment 20% of EAL’s exceed the maximum arrest force (*Fm*) allowed in EN 355. The MCD data provided by manufacturers is not reliable. The data provided by manufacturers are incomplete. It is recommended that quality control for EAL’s be strengthened to ensure that products launched onto the market meet the requirement of EN355. The findings of this study recommended increasing MCD distance longer than that calculated according to EN355, at less than 1 m.

## 1. Introduction

Unfortunately, accidents due to falls from height remain one of the main causes of serious and fatal accidents worldwide [1,2,3,4,5,6,7]. In Spain, according to Employment and Social Security Ministry data, in 2016, falls from height accounted for 42.3% of serious, and 25% of fatal accidents [3]. There is a persistent negative trend, as can be seen by studying accidents that occurred in the United States, where, between 1997 and 2012, accidents due to falls from height rose from 36% in the study by Huang [8], to up to approximately 45% in 2017 [9]. Society is being faced with the need to reverse this trend by providing knowledge of techniques that prevent these accidents and by means of a detailed study of available protection elements [10,11].

To reduce the number of accidents due to falls from height, the correct use of Fall Arrest Systems (FAS) is indispensable [12,13,14,15]. There is a relationship between accidents and failure to use FAS correctly, the improper installation of FAS by the workers is the key to serious and fatal accidents occurring on construction sites [16]. It is, therefore, necessary and essential to understanding safe usage limits for personal protection equipment, as well as having empirical data in order to be able to develop and improve the equipment that provides practical solutions to certain fall risks.

At the construction site, it is common (Figure 1) to give a FAS with EAL to a worker for temporary work at the height [12,13,14,15,16,17], scaffolding work, ladders, mobile elevating work platforms, etc. It is essential that the user knows, without a doubt, the limits of the use of the delivered equipment. The worst scenario in the construction sector is that the anchor point is at the level of the user’s feet and close to the ground level. Figure 2 shows some of these situations. This situation is repeated in the construction of the first floor of any construction, where the distance to the ground floor could be 3 m or less. The red arrows in Figure 2 indicate the free distance to the ground, in all cases less than or equal to 3 m.

A FAS should not allow the user’s body to reach the ground, or another obstacle, during fall arrest. The simplest and commonly used FAS (Figure 1 and Figure 2) consists of a harness, an anchorage point and an EAL. When the FAS is working, a falling mass (*m*) is dropped from an arbitrary height *h* ≤ 2 *L*, where *L* is the EAL length. EALs consist of three parts: the energy absorber (EA); the linear element that determines the length of the equipment, the lanyard (*l*); and the terminal connectors that allow it to be attached to the harness and the anchorage point. Figure 3 shows de worst scenario (*h* = 2 *L*) and the relevant distances in the arresting phenomenon and offers a brief explanation of these devices.

As can be seen in Figure 3, MCD below anchorage is the distance that the user must know to install the anchorage correctly; in other words, it is the minimum required distance below anchorage, so that, in a hypothetical arrest, the accident victim’s body does not hit the ground. Thus, MCD is defined as the obstacle-free distance below the anchorage point. Each manufacturer should provide the user with this information about their equipment, a requirement established in Clause 7.(e) of standard EN355 [18].

The MCD should be:*MCD* ≥ *L* + *S_max_* + *w* + 1(1)
where *L* is the length of the EAL (m); *S_max_* is the EAL maximum extension during the fall (m); *w* is the distance between the harness attachment (chest or back) and the user’s feet (m); *1* is the safety distance equal to one meter. It includes harness extension, body deformation and tall users.

EAL plays the main role of dissipating the kinetic energy generated during a fall from a given height [18,19,20]. This dissipation is achieved by the progressive tearing of the stitching of the EA, this elongation through unstitching plus rope elongation causes the EAL to increase its length significantly, until *S_max_* maximum extension (*S_max_*) has an elastic (*S_e_*) and a plastic (*S_p_*) component Equation (2), both of them are relevant in MCD [21,22].
*S_max_* = *S_e_* + *S_p_*.(2)

Various international standards [18,23,24,25,26] regulate the maximum allowed extension, *X* (m), which is shown in Table 1 together with the rest of the regulatory requirements. Standards say that *X* (m) must be greater than the plastic elongation (*S_p_*) obtained in the test due to the difference between the length at rest after the test and the initial length. The standards methodology [18,23,24,25,26] deem elastic elongation (*S_e_*) to be a negligible value, which, a priori, is not necessarily true. An error in predicting this elastic elongation means that the MCD is underestimated and that the accident victim could reach the floor level.

The maximum length of the EAL before the trial should be less than two meters, as indicated in different standards [18,23,24,25]. This length includes the connectors and is measured from the most extreme points that receive the load (the inside end of the connector).

This study is focused on the worst scenario, that is a fall of a person weighing 100 kg (equipment maximum certified mass) and a Fall Factor (FF) of 2. FF is understood as the height of the fall (*h*) divided by the lanyard length (*L*) from the harness connection to the anchorage tie-off, a definition set out in Clause 2.19 of ANSI/ASSE A10.32-2004 [27].

For this worst scenario, EN 355 [18] standard establishes that the MCD_EN355_ should be Arrest Distance (*H*) plus one meter (standards include a safety distance). *H* is defined as the vertical distance, expressed in meters, between the initial position (start of free fall) and the final position (balance after arrest) of the mobile point of the connection subsystem that supports the load, not counting elongations of the fall arrest harness and its attachment element [28]. The definition and methodology used in EN 355 show that it only takes plastic elongation (*S_p_*) into account.
*MCD_EN355_* ≥ *H* + 1(3)
*H* = *2 L* + *S_p_*.(4)

For both formulas, MCD Equation (1) and MCD_EN355_ Equation (3), to be equivalent, necessarily *L* = *S_e_* + *w*. On the other hand, for the formula proposed by EN 355 to be on the safe side, the condition *L* ≥ *S_e_* + *w* must be fulfilled. Taking into account that *w* is not defined and that *S_e_* is not known, compliance with this equation is not guaranteed. Therefore, it is necessary to establish a method to measure *S_e_* and establish a value for *w*.

Manufacturers provide MCD_MAN_ in their instruction manuals, as shown in Table 2. The distance provided is based on the tests included in the standards, which, as already mentioned, neglect elastic elongation. The MCD value obtained experimentally is compared with the data provided by manufacturers (MCD_MAN_) and with the MCD_EN355_ value obtained by applying standard method Equation (3). These checks are useful for determining the reliability of Equation (3) established in EN 355 [18] and, on the other hand, to ensure that the data provided by the manufacturer guarantees that the user will not hit the ground, or an obstacle, in a hypothetical fall. More important is to determine the reliability of the data provided by the manufacturers and by EN 355.

The key to obtaining the MCD is the initial length of the EAL and its *S_max_* due to impact. A real value should be obtained through tests on each EAL prior to commercialization. Unfortunately, manufacturers do not provide information about this parameter or about tests performed on equipment that they commercialize. Given the lack of information in this regard, both from manufacturers and in the specialized literature [29], this study proposes the performance of experimental tests in order to measure the maximum elongation of the equipment (*S_max_*), plastic plus elastic elongation in the worst scenario that the EAL provides. Therefore, reliably obtain the MCD.

This study is undertaken with the objective of providing empirical data to serve as a basis for the design of the EAL and for adjusting the MCD in the regulatory requirement of EN 355 [18], considering both plastic and elastic deformation, which is not even considered in the international standards. Thus, the aim is to check the safety of equipment available on the market given that there is very little relevant information provided by manufacturers [29,30,31,32], as well as due to the lack of empirical tests on this equipment that provide values for body displacement, MCD from anchorage and plastic and elastic elongations. Therefore, a more particular goal is to give the value of permitted maximum elongation EAL for FAS considering both elastic and plastic deformation of all involved materials.

## 2. Materials and Methods

The test samples were selected to cover the different types of EAL on the market that comply with EN 355 [18]. The choice of samples was made by two technicians with more than 20 years’ experience in the prevention of fall hazards in construction and fire departments, including commonly used equipment. The selection included products from six different manufacturers with international distribution. Ten samples are available, covering different typologies depending on the lanyard.

The MCL_MAN_, the plastic component of EAL elongation (*S_p_*), and the distance from the feet to the attachment point of the user’s harness (*w*) shown in Table 2 are those included in the instruction manuals of the different manufacturers. It should be noted, from the outset, that the values provided by the manufacturers do not fulfill Equation (1).

In cases where the manufacturer does not supply the connectors, the EAL instruction manual was followed in order to choose suitable connectors, steel connectors that comply with EN 362 [33] with a static rupture of 38 kN on the major axis and 10 cm in length on the major axis, were used. Therefore, 20 cm was added to the length declared by the manufacturer in those cases where the connectors were not included, thus obtaining the length declared by the manufacturer, including connectors (*L*).

In Table 2 it can be seen that, in four of the samples selected, the manufacturer does not provide any information regarding plastic elongation of the absorber (*S_p_*), and, in three cases, the data provided corresponds to the maximum allowed by the standard (1.75 m in Europe), which raises doubts about its veracity. On the other hand, it should be noted that there is no unified criterion regarding the value of w, which is required to calculate the MCD Equation (1). Either the data are not provided or a value between 1.5–2 m is applied arbitrarily. Manufacturer B determines that it is the user who takes the value of *w* that he/she sees fit as a function of his/her own height. Given this dispersion of criteria, it is difficult for the user to be clear about MCD, and it is therefore probable that he/she does not know the minimum distance from an obstacle (the ground) at which an anchorage may be installed safely for the use of an FAS with an EAL.

In this study, a method was used that allows the fall arrest process of a worker to be studied. On the one hand, the force-time curves obtained in the dynamic tests were analyzed and, on the other, the images obtained using a High-Speed Camera (HSC) in order to measure the elastic deformation of the EAL precisely. In the dynamic tests, the worst possible scenario was always considered, raising the mass to the maximum allowed by the equipment and then letting it fall, in other words, with a fall with a FF 2; in other words, if the equipment measures 1 m, the maximum fall it allows is 2 m. The study was carried out in the Large Structure Laboratory of the Civil Engineering Department of the University of Alicante. The bench (Figure 4), designed by the authors, was tested successfully in previous research [30,31], and was used for the dynamic tests. The gantry was equipped with a spherical ball joint that allows free oscillation during free fall arrest as established in EN364 [34] and EN355 [18] and meets the requirements of standards for performing tests. The data relating to the force–time curves were obtained at a frequency of 1000 Hz, which implies recording a data point every millisecond. The used load cell is made by HBM [35] type RSCC and it resists a maximum force of 50 kN. The control and data analysis software used was PCSEK [36] from Servosis Testing Machines. This software has a control frequency of up to 40 kHz.

The process was filmed using an HSC, managing to establish the trajectory of the ballast during the arresting phenomenon. The HSC used was a Casio Exilim HS ZR-1000 able to work at 1000 frames per second (FPS) at a spatial resolution of 224 × 65 px. However, as the experiment was indoors, the light intensity was not enough to use that speed. Additionally, having the uncertainty of knowing the actual ballast trajectory, a lower speed was used, in order to have a wider spatial resolution and the best visualization of the process. Therefore, a speed of 480 FPS was used, which has a spatial resolution of 224 × 160 px.

Figure 4 shows a sketch of the test procedure in accordance with EN 355 [18]. First of all, the absorber integrated fastening equipment of the load cell is suspended and a 100 kg mass is suspended from the lower end. The distance *L* between connectors where the load is received is measured using a flexometer. Then, the mass is raised to the maximum allowed by the test sample at a maximum horizontal distance of 300 mm, and then dropped. Adjustable EAL’s (codes V and VI) were tested at the maximum extension length allowed by the equipment.

The force-time curve was recorded and the Maximum Forces (*F_m_*) of the results were obtained directly using the software indicated above. To determine the Average Force (*F_a_*), the values between the first and last time that the value 3.3 kN appears in the curve [24] were taken.

With the mass at rest after the test, the vertical distance between the fastening point of the mass to the fastening point of the equipment to the load cell was measured using a flexometer, which is called final length, *Lf* in expression (5). From Equation (5) *S_p_* is solved, it is substituted in Equation (4) and the Arrest Distance *H* is obtained.
*L_f_* = *L* + *S_p_*.(5)

The image processing provided by the high-speed camera gives *D* corresponding to the vertical distance between the base of the ballast and the ground at the moment of maximum elongation. Knowing the vertical distance between the hook and the ground (4100 mm), the vertical length of the ballast (470 mm) and *S_p_*, it is possible to solve *S_e_* (elastic deformation) of Equation (6) (Figure 5).
*S_e_* = 4100 − 470 − *D* − *L* − *S_p_.*(6)

To obtain *D*, a visualization of the recording of the experiment was made and the frame in which the ballast reaches the minimum distance from the laboratory floor was selected. In the selected frame, the distance in pixels from the ground to the base of the ballast was measured and transformed into the real distance by means of the px/mm ratio obtained for the same recording prior to the test. This ratio is obtained previously using an element of known distance, which is recorded so that its full length appears in the image plane. For the recordings made in this study, the px/mm ratio was 0.5. Once this ratio is known, the desired length in the image is simply measured (in px) and related to its real length. Figure 6 shows the frame corresponding to the maximum elongation for test VIII (with max resolution HSC gives). Some auxiliary lines were drawn on the ground in order to measure the distance to the ground with greater precision. To do so, the fact that the vertical fall of the ballast was located just above the center of a metal plate placed on the floor for safety, was taken into account. Furthermore, it has been assumed that the trajectory of the ballast can be located most of the time and in an approximate way, including in the vertical plane that passes between the gantry supports and which includes the vertical fall of the ballast. I

Prior to the dynamic tests, the samples were acclimatized (temperature and humidity) and their length was then measured. The length of the equipment was obtained directly by measuring the samples suspended from the gantry using a flexometer, ensuring that the equipment was extended, but without load. Initial and final measurements were made, which difference correspond solely to plastic deformation. Elastic deformation can only be measured through image analysis, as described above.

## 3. Results

This section sets out the results, and the regulatory requirement regarding the maximum elongation permitted. Figure 7 shows absorber deployment after the test for each sample. In sample I, the absorber was fully deployed and broke. In the rest of the samples, the absorber did not become fully unstitched, leaving a path in the unstitching area and, therefore, showing an absorption capacity greater than the required quantity for a FF 2 with 100 kg of mass.

### 3.1. Forces

The deformation energy was absorbed with variable force during the arrest process. It was composed of one part of plastic energy generated during the plastic elongation of the EAL, which was normally the greater part, and another smaller part of elastic energy generated during the elastic elongation of the EAL and which was recovered after discharge. Table 3 shows the results obtained for average forces (*F_a_* Exp.) and maximum forces (*F_m_* Exp.). Shown also are the initial length of the equipment (*L*) and the final length (*Lf*) of the same measured at rest before and after impact, respectively.
(7)$$m\cdot g\cdot \left(2\cdot L+S_{max}\right)={F^{\prime }}_{a}\cdot S_{p}+\frac{1}{2}\cdot {F^{\prime }}_{a}\cdot S_{e}.$$

The potential energy at the highest point must be equal to the deformation energy at the lowest point, Equation (7). When operating properly, the theoretical average unstitching force (*F′_a_*) can be solved. The table shows the average theoretical unstitching force (*F′_a_*), calculated according to Equation (8):(8)$${F^{\prime }}_{a}=\frac{m\cdot g\cdot \left(2\cdot L+S_{max}\right)}{\left(S_{p}+\frac{1}{2}\cdot S_{e}\right)},$$
where *m* is operator mass (kg); *g* is the gravitational acceleration (9.81 m/s^2^); *L* is the length of the EAL (m); *F′a* is the average theoretical deployment force (N); *S_p_* is plastic elongation (m); *S_e_* is elastic elongation (m); *S_max_* is total elongation (m).

The maximum forces are obtained from direct reading and the average forces are obtained by analyzing the points of the force-time curve in each test. Test I yields an *F_m_* value above 6 kN, exceeding the injury threshold assumed by EN355 [18]. Test IX is deemed to be at the safety limit. These two items of equipment should not be available on the market because they do not meet the requirements of EN355 [18] and are, therefore, presumed to not comply with Regulation (EU) 2016/425 [37].

In light of the results obtained, the two EAL’s corresponding to tests I and IX can be eliminated from the analysis group (because they do not meet the regulatory requirement [18] of not achieving a maximum arrest force of 6 kN). In order to evaluate if differences exist between the results of test I and IX and the rest of the data, the Kruskal–Wallis test can be applied [38]. The Kruskal–Wallis test is a non-parametric method to evaluating when two or more samples provide from the same distribution. The null hypothesis of the test is that medians of all the groups are equals. Table 4 presents the statistical value and *p*-value (significance level) for the Kruskal–Wallis test, and it can be concluded that significant differences are present in the samples. Notice that a significance level of 0.03671 is the probability of obtaining test results under the assumption that the null hypothesis is correct, and this value is less than the standard significance level of 0.05.

A non-parametric test had to be applied to validate the hypothesis that the *F_m_* is, for example, 6.5 or 4.5 kN. The descriptive statistics for *F_a_* (8 samples) are: minimum, 4.127, first quartile, 4.207, median, 4.281, mean, 4.474, third quartile, 4.694, and maximum, 5.177 N. The Wilconxon test [39] can be used as an alternative to the parametric Student’s *t*-test. The Wilconxon test allows us to determine whether the median is equal to a known theoretical value. Table 5 shows the results obtained by applying the Wilcoxon test to our data, for the different null hypotheses of *F_m_,* from which it can be concluded that the median *F_m_* is lower than 5 kN with a significance of 5%. This follows from the *p*-value (0.0017) in this case, being the probability of obtaining test results at least as extreme as the observed ones. Therefore, the probability that the *F_m_* is 5 or higher is 0.0117. The text continues here.

To compare the *F_a_* Exp. data obtained in the laboratory with the Theoretical *F′_a_* (Equation (8)), the difference between them has been taken, since the data were two measurements from the same test and see whether there are significant differences. Table 6 shows the data that were the object of the study.

The descriptive statistics for *F′_a_*–*F_a_* Exp. are: minimum, −0.0540, first quartile, 0.2972, median, 0.3345, mean, 0.4228, third quartile, 0.4228, and maximum, 1.2370 N. In order to analyze this sample, it is interesting to know whether the distribution of the values of the mean forces obtained follow a normal distribution. Figure 8 compares the probability distributions of our sample with the normal distribution. Now, let us see if it is acceptable that the sample comes from a normal population.

The Shapiro–Wilk normality test gives us a test to contrast whether a sample comes from a normal distribution with a *p*-value = 0.1487. By applying the non-parametric Wilcoxon test, it is obtained: dif. V = 1, *p*-value = 0.001953. Both the parametric *t*-test and the non-parametric Wilcoxon test provide the same results. The difference between *F′_a_–F_a_* Exp. is significantly greater than zero, with a significance value less than 0.01.

The only case in which the difference is not negative does not detract from the significance of the test, it strengthens the idea that the EAL from Test I should be withdrawn from the market since it provides a value far removed from the rest of the equipment tested.

Next, some statistical values obtained from the samples are presented. Although the use of parametric statistics, in order to make any statistical inference, requires a number of samples significantly greater than those present in this study, it may be convenient to present these values to illustrate the differences observed in the results of the experiments. These statistical values should be considered with caution.

Pearson’s linear correlation coefficient is 0.797, if all data are considered, and rises to 0.953 if only the eight pairs of data are considered. In either case, the estimation by linear regression of the theoretical force through the experimental force provides an adjustment coefficient of 1.12. In other words, on the side of safety, the theoretical force was overestimated by 12% based on the experimental force calculated.

It is confirmed that there is a linear relationship between *F′_a_* and *F_a_* Exp. By quantifying the difference between *F′_a_* and *F_a_* Exp., it is concluded that *F′_a_* is valid for calculating *F_a_* Exp. *F′_a_* explains 90% of the variability present in F_a_ Exp. (Linear regression of *F′_a_* and *F_a_* Exp. R-squared: 0.9077; F-statistic: 58.98; *p*-value: 0.000255).

The following graph (Figure 9) represents *F_m_*, *F_a_* Exp. and the free fall height in each case, it is confirmed that there is no linearity between the two variables and that there are manufacturers that achieve lower impact forces at higher fall heights. The data have been ordered by free fall height for easier visual understanding.

### 3.2. Elongations

Plastic elongation (*S_p_* in Table 7) is found in all cases below the 1.75 m, maximum allowable extension (*X*), required by EN 355 [18]. In the majority of the equipment, elastic deformation accounts for approximately 18% of the total, significantly less than plastic deformation; therefore, a test and regulatory requirement based solely on plastic elongation seem, at first glance, to be adequate.

Notwithstanding, it is worth noting that new, highly elastic designs, like the one used for test IX (Table 7), with an elastic elongation of around 70% of the total, are likely to meet the regulatory requirement (<1750 mm). In fact, have an elongation (highly elastic) such that the operator would reach an obstacle below (the regulatory requirement only considers plastic elongation). In this specific case (test IX), 70% of the elongation that occurs at the moment of maximum extension, is disregarded.

The elongation as a function of free fall, is shown in Figure 10. It should be noted that the objective is equipment that minimizes force and elongation at any fall height. The equipment with the greatest free fall (test X) was fourth in terms of least elongation. It could be assumed that this penalizes the arrest force, but by looking at Figure 6, it can be seen that the force value was below 5 kN.

The relationship that exists between free fall and *S_max_* is linear, the correlation between the two measurements being 0.85 (10 samples). The fall distance (free fall) explaining 72% of the variability observed in the *S_max_*. The adjustment coefficient between the two sets of data was 0.333 (the slope of the regression line), therefore, it can be concluded that the *S_max_* represents a third of the fall distance.

The maximum opening (Figure 11) obtained was 1.5 m, 14% less than the maximum allowed by the standard. For absorbers with large elastic deformations (test IX), the use of the arrest distance defined in the standard is very far removed from the real value of the arrest distance. For the rest of the equipment, this variation does not exceed 17%.

### 3.3. Minimum Clearance Distance below Anchorage (MCD)

In this section, the results obtained from the calculation of the MCD below anchorage based on the measurement of the elastic elongation obtained from the analysis of the images obtained from the HSC recordings, are shown. The results obtained are compared with the data provided by the manufacturers and with the results obtained by applying the provisions of EN 355 [18].

Table 8 presents these data, MCD is the real required distance obtained using the HSC, MCD_MAN_ is the required distance according to the equipment manufacturer and MCDen355 is the required distance calculated according to EN 355.

To calculate the MCD and the MCD_EN355_, the real length (*L*) of the EAL’s measured in the laboratory was taken into account, not the one declared by the manufacturer. In order to unify criteria regarding the distance from the feet of an operator to the harness attachment point (*w*), the anthropometric data of the Spanish working population [40] published by the National Institute for Health and Safety at Work (INSST), were consulted. The parameter *w* was taken as the vertical distance from the support surface of the feet to the highest point of the acromion for a 99th percentile. The highest point of the acromion corresponds to the attachment point of the harness. Therefore, *w* is 1588 mm, this value being in line with Small (2011) [29]. The results from following the indications of EN 355 [18] and calculating MCD_EN355_ according to Equation (2) are shown in Table 8.

Dong (2017) [16] considers an MCD below anchorage of 5.33 m and Epp (2017) [41] states that a distance less than 4.57 m from the anchorage to the ground renders the equipment ineffective. These authors propose a single MCD value. This study, by contrast, shows that the MCD has a very wide range of values. Specifically, the items of equipment studied need between 3.9 m and 6.1 m of MCD below anchorage. This implies that EALs can be purchased on the market, which are capable of arresting a fall below 4 m safely while others require more than 6 m. To optimize the use of an item of equipment, an individual study of its characteristics is required. It is, therefore, of vital importance that the applicable regulations oblige each manufacturer to provide the specific MCD for their product.

There are indeed differences between MCD and MCD_MAN_. The non-parametric Wilkinson test supports the hypothesis that manufacturers are conservative, in other words, MCD < MCD_MAN_, with a significance level below 5%.

Of particular concern is case I when comparing the data provided by the manufacturers (MCD_MAN_) with the distance calculated using the HSC (MCD) it is demonstrated that the distance provided by the manufacturer is insufficient to arrest the fall without the operator hitting the ground.

It is also significant to see that the data provided by manufacturers are, for some items of equipment, not so far removed from reality (below 300 mm), while, for others, however, the difference exceeds 1300 mm, the manufacturer appearing to be excessively conservative, in this case, from a safety perspective.

For FF 2 falls, EN 355 [18] establishes that clearance below anchorage should be calculated according to Equation (2). When this distance, MCD_EN355_, is calculated and compared with the real one, MCD, obtained in the laboratory using an HSC as described in Equation (1), it can be seen that, in six out of the ten cases studied, the standard is removed from reality, putting the user of the equipment at risk. The standard calculation displays a difference, from a lack of safety perspective, of up to 928 mm, which would put the user at risk of hitting the ground or another obstacle. It is necessary that the standard reformulates the way of calculating MCD for FF 2 falls with a mass of 100 kg. In a practical way, it can be established that one meter must be added to the distance obtained in the application of EN 355. Therefore, MCD_EN355_ + 1 = MCD.

This same item of equipment I, is doubly dangerous since it also fails to meet the regulatory requirement of keeping the *F_m_* below 6 kN. For the equipment of tests II and VII, safety depends on the user installing the anchorage taking *w* (a distance between the harness attached to the user’s feet greater than 1363 mm and 903 mm, respectively).

## 4. Discussion

In principle, the main limitation of the study is the relatively low number of tests performed, 10. Notwithstanding, previous studies have shown that the production of this equipment is highly homogeneous [19,42]. Likewise, personal protective equipment (PPE) in general, and EAL’s in particular, are subject to strict production regulations, Regulation EU 2016/425 [37] requires quality control that guarantees homogeneity in the manufacture of this equipment and a small sample is, therefore, acceptable for the number of tests to be performed.

The study was carried out on EAL’s with absorbers according to EN 355 [18], for FF 2 falls. They are similar to the EA’s of AS/NZS 1891.1:2007 [25], type E6 EA’s of Z259.11 [26] and Z359.13 [24] and type 2 EA’s of ISO 10333-2 [23]. Minor variations may mean that the results obtained here may not be directly applicable to other certified equipment according to other standards.

It is remarkable that 20% of the EAL’s studied exceed the maximum force allowed, evidence that the controls established have not been sufficient or effective. The authors recommend an increase in the control of the manufacture and commercialization of EAL’s.

The *F_a_* was calculated based on clause 4.1.102.2 of ANSI/ASSP Z359.13 [24] which establishes that the values between 2.2 kN be taken the first and second time they appear on the force-time graph, these values being for type E4 [19] low capacity absorbers, used for FF 1 falls and an *F_m_* below 4 kN [43]. In the case that concerns us, for FF 2 falls and assuming certain linearity, for *F_m_* (6 kN), values above 3.3 kN must be taken. The *F_a_* range thus obtained spans from 3.7 to 4.5 kN, higher than to be expected from those of the American Standards, which are between 2.67 and 3.6 kN described in Z259.16 and Z359.6 referring to E4. This is similar to the 3.2 to 4.7 kN range calculated by Goh [19] for absorbers according to AS/NZS 1891.1:2007 [25]. The *F_a_* calculated based on the principles of energy conservation differ by 12% from the experimental one. This could be influenced by the effects of heterogeneous tearing and by the absorption of energy of other components such as the tapes and connectors of the EAL’s, which has a slight effect on the Energy Conservation equation Equation (2) from a safety perspective. Therefore, in the absence of data provided by manufacturers, it can be used to calculate the *F_a_.* The *F_a_* is required, for example, to estimate the opening of an EA and to determine the obstacle-free distance below the anchoring in a FAS Equation (8).

Standard EN 355 [18] disregards plastic elongation but applies a high safety coefficient (Table 9) to assume the 20% elongation that is ignored. The requirement of 1750 mm of maximum elongation allowed results in an increase above the real elongation (% *S_max_*) that ranges from 17 to 274%, which is an exaggeratedly wide range.

In order to unify criteria, regarding the distance from the operator’s feet to the harness attachment point (*w*), the anthropometric data of the Spanish working population were consulted. For application in other countries, the endemic data from each specific country should be consulted.

It would be of great value to study the EAL’s with different masses to have evidence of their behavior in rescue situations whereby two operators are hanging from one item of equipment. A study with greater masses would enable it to be determined how safe the equipment is in the case of workers who weigh more than 100 kg.

## 5. Conclusions

After analyzing the results obtained, it can be concluded that most manufacturers are conservative and their EAL’s have oversized unstitching paths. Only one EAL becomes completely unstitched, the rest do not reach the limit of their absorption capacity.

The difference between the initial lengths (before the fall) of the equipment declared by the manufacturers and their real length *L* is significant (>10%) in 2 of the 10 samples studied. The difference found errors on the side of safety since the real length is less than that declared. Four cases were found in which the equipment was longer than the length declared by the manufacturer (up to 3.5%).

The EAL’s corresponding to tests I and IX should not be on the market since the *F_m_* exceeds the threshold for injury established at 6 kN. For the rest of the equipment, it is around 5 kN or below. The forces obtained theoretically Equation (8) are 12% higher than those obtained experimentally. Therefore, Equation (8) is a good tool for calculating the *F_a_* with EAL’s certified under EN 355 [18].

The average value of the measured force (*F_a_*) ranges from 3.7 to 4.5 kN, somewhat higher than in previous studies [19,38,41,43]. As expected, the fall height and the arrest force do not present linearity, in line with other studies [12,30,31,32]. At the same FF, the intrinsic characteristics (type and location of stitching) of each EAL have a significant influence on the arrest force value.

The maximum elongation of 1.75 m established by EN 355 [18] could be reduced in light of the results obtained in this investigation. The maximum value obtained was 1.24 m. Therefore, the requirement could be set at around 29% lower than the currently established value of 1.75 m. This would be without the need to raise the force received by the operator in the arrest, which, as it’s been shown here, is around 5 kN.

The maximum elongation allowed disregards up to 70% of the elongation of the EAL (Test IX). It could be the case that there is an EAL on the market satisfying EN 355 [18] but after an accident, the user hits the ground due to elastic elongation. As already mentioned, the regulatory requirement only considers plastic deformation. It is worrying that the method used in EN 355 [18] to determine elongation could disregard 70% of the value it is intended to measure.

Notice that the method outlined in this study is easy and inexpensive and capable of providing the total, elastic and plastic elongation of an EAL. According to the authors, it could be interesting to include this method in the test methods described in standards EN355 [18], ISO 10333-2 [23], ANSI/ASSE Z359.13 [24], AS/NZS 1891 [25] and CSA Z259.11.17 [26]. Results here obtained can be applied to any activity in which the use of FAL should be used, such as mountain rescuing, firefighting or any work carried out at heigh. Elastic deformation measurements with HSC can be safely carried out successfully in any sector in which it is required to measure these deformations. Using an HSC would represent an advance in the determination of EAL openings, which would be useful for rigorously determining the MCD below anchorage. Optimization of this distance would make EAL’s more versatile. This will be very useful for manufacturers to optimize their products by modifying the length of their systems and their recommendations of use in order to guaranty worker safety. Furthermore, in the future, the strength of the used ropes and connections for the actual material used on those systems should be also studied for dynamic conditions, including a critical point of view of the present codes related to those issues.

On the other hand, analysis of the data collected from the manufacturers compared to the results from the laboratory shows that the MCD below anchorage values provided by the manufacturers are not reliable. In one of the cases (test I), the manufacturer is 0.5 m short with respect to the value obtained in the laboratory. In this case, a user that trusts the information provided by the manufacturer could, in the event of a fall, suffer very serious injuries since they could be at risk of hitting the ground in a hypothetical fall. In the rest of the cases, the manufacturers err on the side of conservatism, a difference of more than 1 m from a safety perspective with respect to the distance calculated by the authors, which already includes an Sf of 1 m.

In 60% of cases, the real safety distance required calculated experimentally, MCD, is greater than MCD_EN355_ obtained according to EN 355 [18], by applying Equation (2). The authors propose not using this formula based solely on plastic elongation and the length of the equipment.

Neither EN 355 [18] nor the manufacturers provide a clear criterion regarding the *w* length between the anchorage point of the harness and the user’s feet (values that differ by 1.5 m to 2 m between the various manufacturers). Currently, this datum is known and has been studied in various anthropometric papers, the distance between a person’s feet and shoulders being a useful value; the authors, therefore, recommend that the criterion be unified, and they propose, for Spain, a *w* value of 1.588 m as the distance between the anchorage point of the harness and the user’s feet, given the results shown in this paper.

The need to demand more complete tests in the certification process is evident and for these results to be included in the instruction manual that the manufacturer provides to the user. Specifically, the manual should provide information about: *F_m_*, *F_a_*, *S_max_*, *S_p_* and *S_e_.* With these data, a prevention technician charged with procuring the equipment can assess the purchase according to the safety and versatility offered by the equipment. An Engineer could perform the calculations required to determine the clearance distance below anchorage. At present, the data offered by the manufacturers do not allow a choice to be made based on the safety of the equipment.

This study proposes that production controls for this type of equipment be raised by the European legislation to prevent non-conforming equipment from reaching the market.

Taking into account the findings of this study, in a scenario of m = 100 kg with FF 2, it is recommended that users increase the installation distance of the anchor by one meter over the value obtained from applying EN 355 and by 0.6 m the distance provided by the manufacturer.

## Figures and Tables

**Figure 1 ijerph-18-05823-f001:**
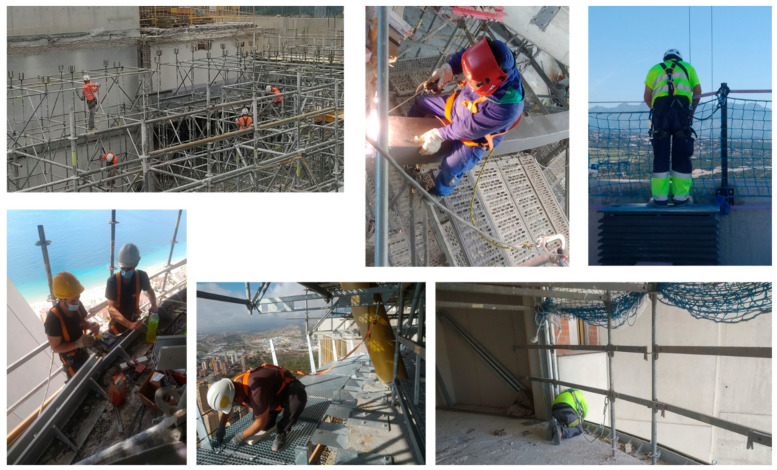
Temporary work at height with FAS with EAL.

**Figure 2 ijerph-18-05823-f002:**
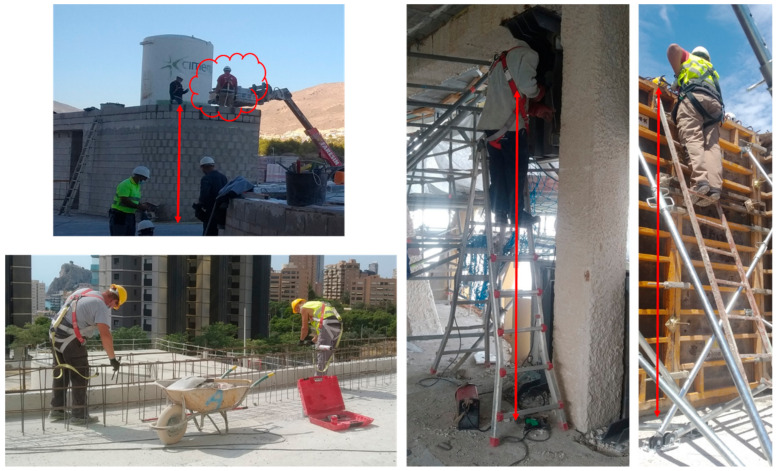
FAS with EAL near the ground level.

**Figure 3 ijerph-18-05823-f003:**
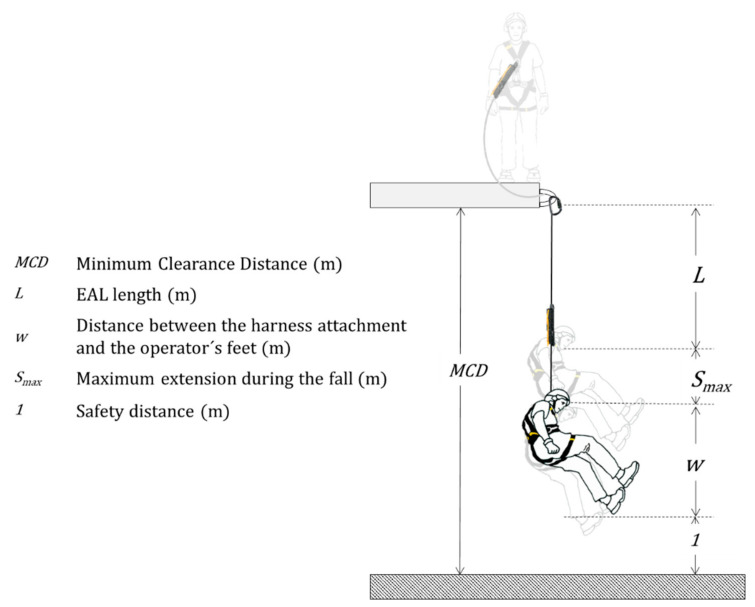
Relevant distances in FAS.

**Figure 4 ijerph-18-05823-f004:**
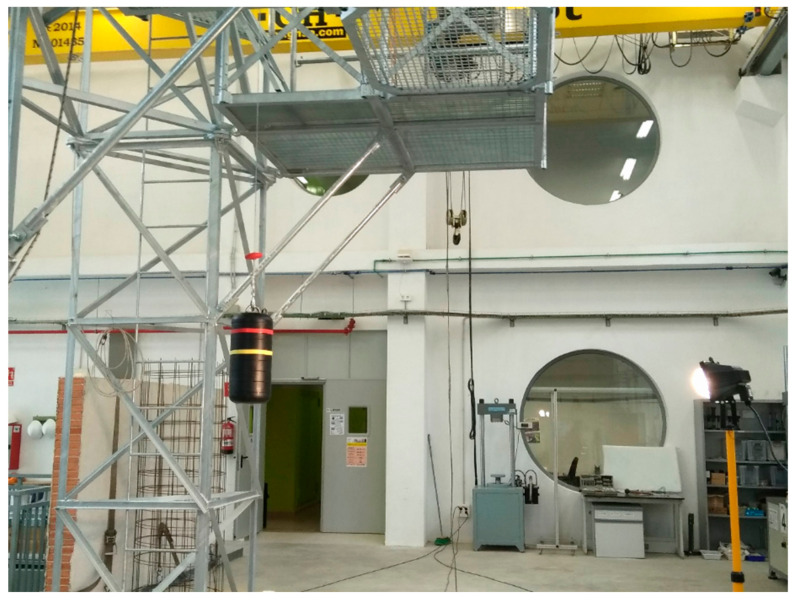
Test bench and mass.

**Figure 5 ijerph-18-05823-f005:**
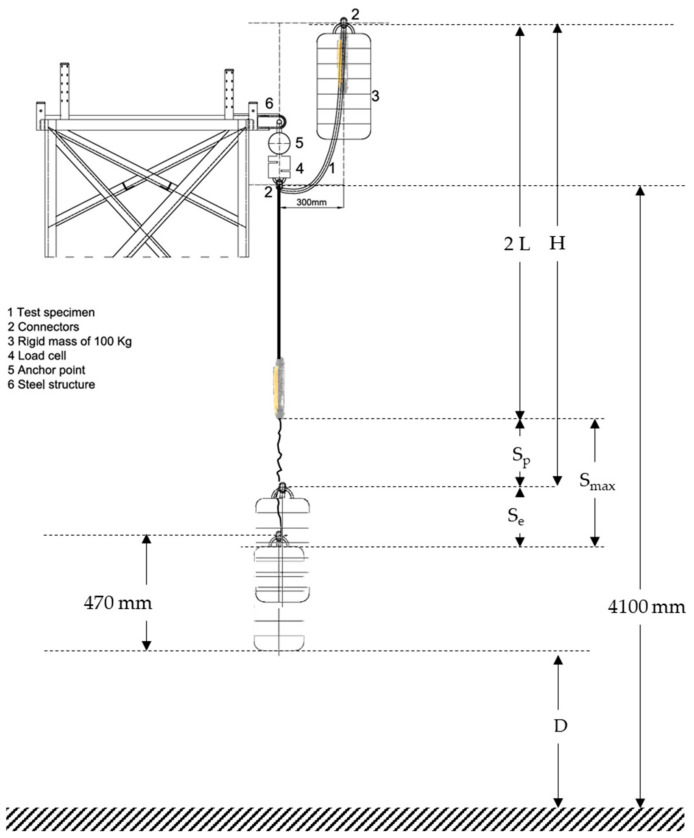
Dynamic behavior testing procedure.

**Figure 6 ijerph-18-05823-f006:**
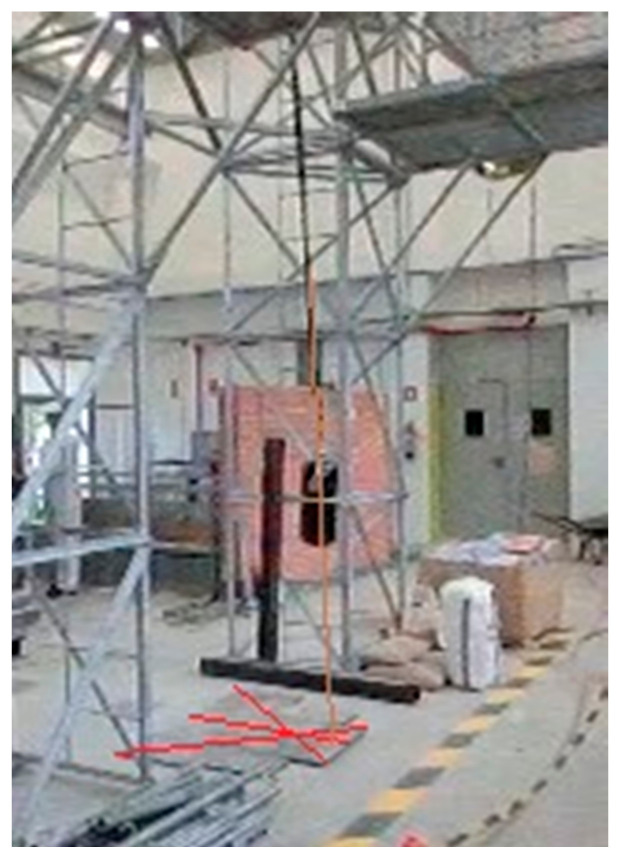
Frame corresponding to the maximum stretch moment for experiment VIII. Some auxiliary lines in the ground help to locate the point to measure the distance from it to the bottom of the ballast.

**Figure 7 ijerph-18-05823-f007:**
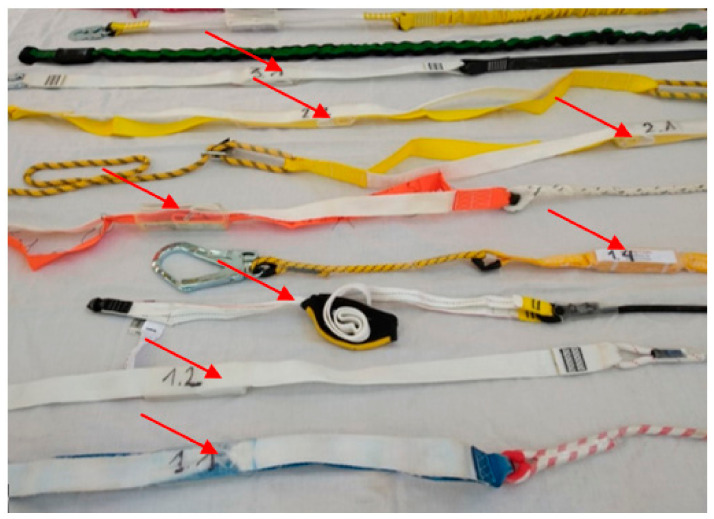
Samples after dynamic behavior testing.

**Figure 8 ijerph-18-05823-f008:**
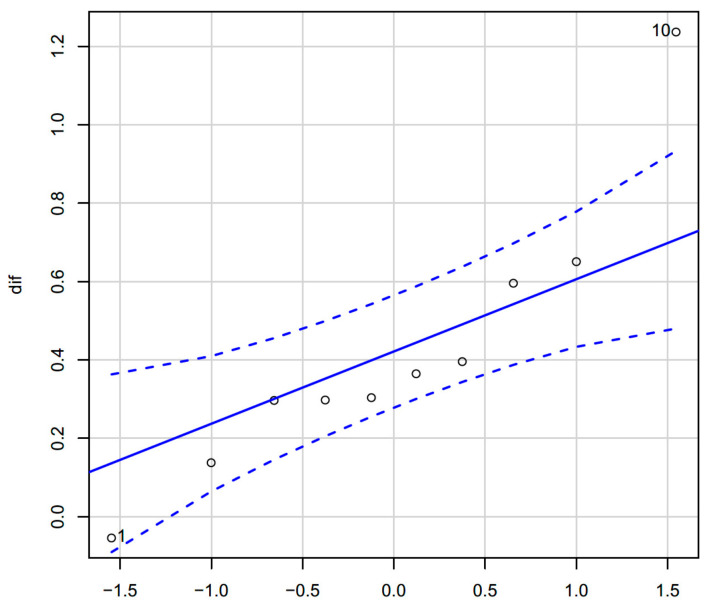
Normal Q–Q plot for *F′_a_–F_a_* Exp.

**Figure 9 ijerph-18-05823-f009:**
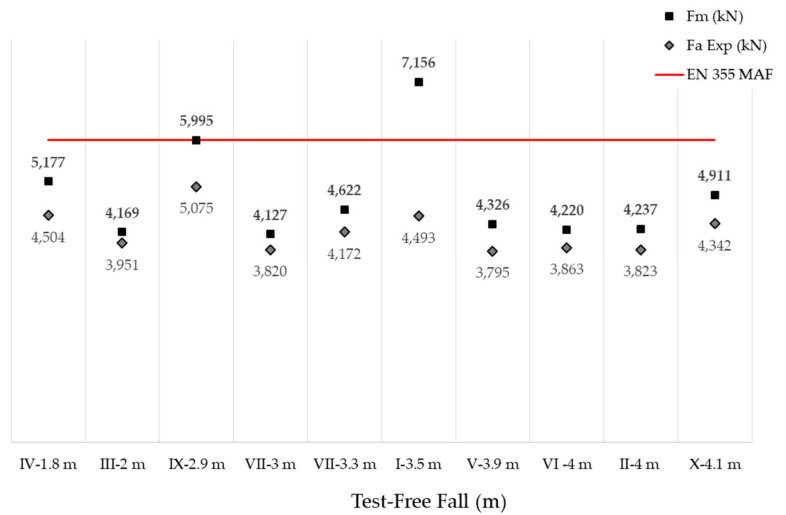
Freefall vs. maximum arrest force (*F_m_*) and average deployment force.

**Figure 10 ijerph-18-05823-f010:**
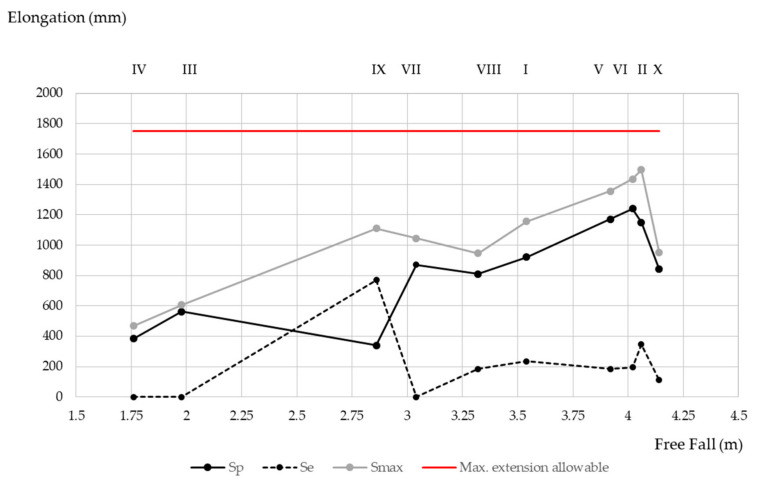
Elongation vs. free fall distance.

**Figure 11 ijerph-18-05823-f011:**
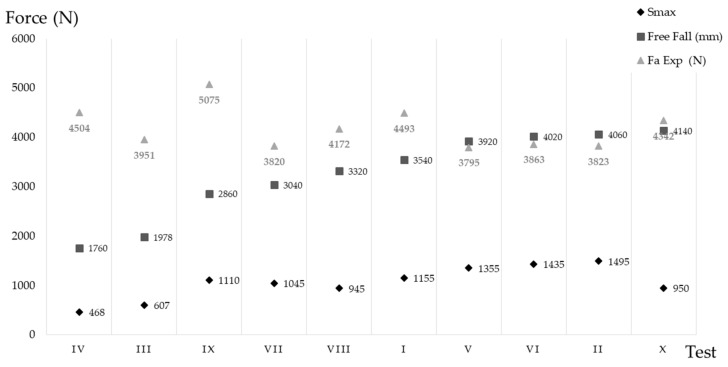
*Fa* Exp, free fall distance and elongation.

**Table 1 ijerph-18-05823-t001:** International standards requirements.

Standard	Test Mass *m* (kg)	Free Fall Distance *h* (m)	*X* (m)	*F_m_* (kN)
ISO 10333-2:2000 TYPE1	100	1.8	1.2	4
NSI/ASSE Z359.13-2013	128 ^1^	1.83	1.2	8
ANSI/ASSE Z359.13-2013	128 ^1^	3.66	1.5	8
ISO 10333-2:2000 TYPE 2	100	4	1.75	6
AS/NZS 1891.1 2007	100	4	1.75	6
Z259.11.17	manufacturer	manufacturer	0.7~0.95 (*X_MAN_* ^2^)	8
EN 355:2002	100	4	1.75	6

^1^ Conversion factor 1.1 is being used comparing rigid test weight to the human body (140 kg); ^2^ Maximum elongation that manufacturer declares.

**Table 2 ijerph-18-05823-t002:** Samples technical specifications according to the manufacturer’s instruction manual.

Code	Type	Manufacturer	Connectors	*L* (m)	*MCD_MAN_* (m)	*S_P_* (m)	*w* (m)
I	Rope + EA	A	No	2	5	1.75	1.5
II	Rope + EA	B	Yes	2	4.75 + *w* ^2^	1.75	-
III	Rope + EA	C	No	1	4.4	-	-
IV	Rope + EA	D	Yes	1.1	4.2	-	-
V	Adjustable rope + EA	D	Yes	2	6.2	1.2	2
VI	Adjustable rope + EA	D	Yes	2	6.2	1.2	2
VII	Webbing + EA	B	Yes	1.5	4.25 + *w*	1.75	-
VIII	Elastic webbing	F	No	1.7	6.5	-	-
IX	Elastic webbing + EA	D	Yes	2	6.2	1.2	2
X	Wire + EA	D	No	2	3 ^1^	-	-

^1^ The manufacturer of this equipment contemplates a use as a fall arrest device limited by the installation system to factor 1 falls (FF = 1). However, it has been tested under the same conditions as the rest of the devices. ^2^ *w* is the distance between harness attachment and the user’s feet.

**Table 3 ijerph-18-05823-t003:** Test results and *F′**a*.

Test	*L* (mm)	*Lf* (mm)	*F_m_* Exp. (kN)	*F_a_* Exp. (kN)	*F′_a_* (kN)
I	1770	2690	7156	4493	4439
II	2030	3180	4237	3823	4121
III	989	1152	4169	3951	4347
IV	880	1263	5177	4504	5155
V	1960	3130	4326	3795	4099
VI	2010	3250	4220	3863	4001
VII	1520	2390	4127	3820	4185
VIII	1660	2470	4622	4172	4768
IX	1430	1770	5995	5075	5372
X	2070	2910	4911	4342	5579

**Table 4 ijerph-18-05823-t004:** Kruskal–Wallis.

	Statistic	*p*-Value
Kruskal–Wallis	4.3636	0.03671

**Table 5 ijerph-18-05823-t005:** Results (8 samples) of the Wilcoxon test for *F_m_*.

Hypothesized Median	Alternative Hypothesis	Statistic	*p*-Value	Value Null Hypothesized at 5% Sig	Value Null Hypothesized at 1% Sig
6	<6	0	0.0039	Reject	Reject
5	<5	2	0.0117	Reject	Accept
4.5	<4.5	16	0.4219	Accept	Accept

**Table 6 ijerph-18-05823-t006:** *F′_a_*–*F_a_* Exp.

Test	I	II	III	IV	V	VI	VII	VIII	IX	X
*F_a_* Exp	4.493	3.823	3.951	4.504	3.795	3.863	3.820	4.172	5.075	4.342
*F′_a_*	4.439	4.121	4.347	5.155	4.099	4.001	4.185	4.768	5.372	5.579
*F′_a_*–*F_a_* Exp	−0.054	0.298	0.396	0.651	0.304	0.138	0.365	0.596	0.297	1.237

**Table 7 ijerph-18-05823-t007:** Elongations.

Test	*D* (mm)	*S_p_* (mm)	*S_e_* (mm)	*S_max_* (mm)	*X* (mm)	*S_e_*/*S_max_* × 100
I	705	920	235	1155	1750	−20.35
II	105	1150	345	1495	1750	−23.08
III	2033	563	45	607	1750	−7.25
IV	2282	383	85	468	1750	−18.16
V	315	1170	185	1355	1750	−13.65
VI	185	1240	195	1435	1750	−13.59
VII	1065	870	175	1045	1750	−16.75
VIII	1025	810	135	945	1750	−14.29
IX	1090	340	770	1110	1750	−69.37
X	610	840	110	950	1750	−11.58

**Table 8 ijerph-18-05823-t008:** Minimum clearance distance (mm).

Test	MCD	MCD_MAN_	MCD_EN355_	Difference MCD_MAN_−MCD	Difference MCD_EN355_−MCD
I	5513	5000	5460	−513	−53
II	6113	4750 + *w*	6210	*w* ≥ 1363	97
III	4184	4400	3541	216	−643
IV	3936	4200	3143	264	−793
V	5903	6200	6090	297	187
VI	6033	6200	6260	167	227
VII	5153	4250 + *w*	4910	*w* ≥ 903	−243
VIII	5193	6500	5130	1307	−63
IX	5128	6200	4200	1072	−928
X	5608	6000	5980	392	372

**Table 9 ijerph-18-05823-t009:** Experimental elongation vs. EN 355 requirement.

Test	*S_p_*	*S_max_*	EN 355	% *S_p_*	% *S_max_*
I	920	1155	1750	90	52
II	1150	1495	1750	52	17
III	563	607	1750	211	188
IV	383	468	1750	357	274
V	1170	1355	1750	50	29
VI	1240	1435	1750	41	22
VII	870	1045	1750	101	67
VIII	810	945	1750	116	85
IX	340	1110	1750	415	58
X	840	950	1750	108	84

## Data Availability

Some or all data, models, or code that support the findings of this study are available from the corresponding author upon reasonable request.

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
