# Peer review of "Minimum Clearance Distance in Fall Arrest Systems with Energy Absorber Lanyards"

_ijerph, 2021, doi:10.3390/ijerph18115823_

Round 1
Reviewer 1 Report
I propose to specify the purpose of the described research more precisely (the goal is given at the end of the introduction, but too general). In my opinion, the goal is also to give the value of "permitted maximum elongation" EAL for FAS.
I have some editorial comments:
- In summary: instead of "The method described in European Standard (EN) 355 underestimates elongation by up to 70% for some equipment" I propose "The method described in EN 355: 2002 underestimates elongation by up to 70% for some equipment".
- Notations "L", "Smax" etc. in the text of the manuscript should be written in italics (pages 4 and following). Units (e.g. length) should be written in regular font (Table 1).
- On page 7, incorrectly says "1000 frames per second (fps) at a spatial resolution of 224 x 65px.", It should read "1000 frames per second (FPS) at a spatial resolution of 224 × 65 px.". These types of errors (FPS, bad sign "x" instead of "×" and space between the number and the unit) appear later in the text.
- Figure 6 is too low resolution.
- The sentence "This section sets out the results, and the regulatory requirement regarding the maximum elongation permitted is analyzed." on page 9 is redundant.
- Page 17: "The range of Fa’s obtained spans from 3.7 to 4.5 kN" or "The measured value of the force F was from 3.7 to 4.5 kN"?
Overall: a very interesting and important topic of the article. Useful results.
Reviewer 2 Report
This is an interesting work and could be useful in practices. I have some suggestions for minor revision:
(1) You are suggested to avoid using "I", "we", "you", etc., in technical writing.
(2) Please double check Table 2. I do not know the difference between V and VI. Please provide more information about the different EAL such as photos. Also, add a note to explain the meanings of w and x in the table.
(3) What did you want to show in Fig. 7? Please elaborate in text and mark in the figure.
(4) Please clearly explain how you did the measurement for forces and lengths.
Reviewer 3 Report
The main concern in the study is the sample size of 8 which is too small for making any statistical inferences.
Even for doing the t-test for hypothesis testing/ mean comparison, at least between 10-15 samples are required.
Apart from this concern, the manuscript can benefit from some modification listed in the following: The literature on the application of similar systems and the statistics regarding the record of accidents with/without using this system should be included in the literature review.
The methodology should include a more detailed analysis of the statistical hypothesis testing, and the tables explaining the results of the hypothesis test should have proper criteria for rejecting/accepting the hypothesis.
In the conclusion section, the future direction of this research should be discussed as well as other sectors that could use such a system.
Furthermore, the details of the application of this research for manufacturers' should be discussed as it was mentioned as ne of the motivations for the study.
Round 2
Reviewer 3 Report
Thanks for addressing the concerns.
Author Response
Dear Reviewer
Thanks for your revision.